

# CHEEREIO 1.0: a versatile and user-friendly ensemble-based chemical data assimilation and emissions inversion platform for the GEOS-Chem chemical transport model

Drew C. Pendergrass[1], Daniel J. Jacob[1], Hannah Nesser[1], Daniel J. Varon[1], Melissa Sulprizio[1], Kazuyuki Miyazaki[2], and Kevin W. Bowman[2].

[1] School of Engineering and Applied Sciences, Harvard University, Cambridge, MA, USA
[2] NASA Jet Propulsion Laboratory, Pasadena, CA, USA

*Correspondence to*: Drew C. Pendergrass (pendergrass@g.harvard.edu)

**Abstract**. We present a versatile, powerful, and user-friendly chemical data assimilation toolkit for simultaneously optimizing emissions and concentrations of chemical species based on atmospheric observations from satellites or suborbital platforms. The CHemistry and Emissions REanalysis Interface with Observations (CHEEREIO) exploits the GEOS-Chem chemical transport model and a localized ensemble transform Kalman filter algorithm (LETKF) to determine the Bayesian optimal (posterior) emissions and/or concentrations of a set of species based on observations and prior information, using an easy-to-modify configuration file with minimal changes to the GEOS-Chem or LETKF code base. The LETKF algorithm readily allows for non-linear chemistry and produces flow-dependent posterior error covariances from the ensemble simulation spread. The object-oriented Python-based design of CHEEREIO allows users to easily add new observation operators such as for satellites. CHEEREIO takes advantage of the HEMCO modular structure of input data management in GEOS-Chem to update emissions from the assimilation process independently from the GEOS-Chem code. It can seamlessly support GEOS-Chem version updates and is adaptable to other chemical transport models with similar modular input data structure. A postprocessing suite combines ensemble output into consolidated NetCDF files and supports a wide variety of diagnostic data and visualizations. We demonstrate CHEEREIO's capabilities with an out-of-the-box application, assimilating global methane emissions and concentrations at weekly temporal resolution and 2°x2.5° spatial resolution for 2019 using TROPOMI satellite observations. CHEEREIO achieves a 50-fold improvement in computational performance compared to the equivalent analytical inversion of TROPOMI observations.



## 1 Introduction

Data assimilation is a field of applied mathematics that studies the most probable combination of a

physical model and observational data to define the state of a system. Many modern data assimilation

algorithms have been motivated by problems in numerical weather prediction [*Kalnay* 2003], and the

field has more recently expanded to address problems in atmospheric chemistry [*Elbern and Schmidt*,

2001; *Kahnert* 2008; *Bocquet et al*, 2015]. The physical model for chemical data assimilation is a

chemical transport model (CTM) that simulates the 3-D fields of species concentrations by solving the

corresponding continuity equations on an Eulerian grid [*Brasseur and Jacob*, 2017]. With the advent of

satellite constellations measuring atmospheric composition together with increasingly dense networks of

surface observations, chemical data assimilation is now commonly used to quantify emissions [*Miyazaki*

*et al.,* 2017; *Jiang et al.*, 2018; *Qu et al.*, 2019a], to construct 3-D concentration fields for chemical

reanalyses and forecasts [*Miyazaki et al.*, 2015; 2020; *Flemming et al.,* 2015; *Ma et al.*, 2019], and to

diagnose CTM biases [*Emili et al.*, 2014; *Stanevich et al.,* 2021]. The use of chemical data assimilation

to quantify emissions is commonly referred to as an inversion in the atmospheric chemistry community.

Most data assimilation algorithms involve the optimization of a Bayesian scalar cost function $J(\pmb{x})$

assuming Gaussian error probability density functions (pdfs) [*Brasseur and Jacob*, 2017]:

$$J(\pmb{x}) = \left(\pmb{x} - \pmb{x}^{\pmb{b}}\right)^{T} (\pmb{P}^{\pmb{b}})^{-\pmb{1}} \left(\pmb{x} - \pmb{x}^{\pmb{b}}\right) + \left(\pmb{y} - H(\pmb{x})\right)^{T} \pmb{R}^{-\pmb{1}} (\pmb{y} - H(\pmb{x})) \qquad \textbf{(1)}$$

Here $\pmb{x}$ is the state vector to be optimized (consisting of emissions and/or concentrations), $\pmb{x}^{\pmb{b}}$ is the initial

physical model prediction of the state vector, $\pmb{P}^{\pmb{b}}$ is the prior (also called background or forecast) error

covariance matrix of the model prediction, $\pmb{y}$ is the suite of observed atmospheric concentrations arranged

as a vector, $H(\cdot)$ is an observation operator that transforms the state vector $\pmb{x}$ from the state space to the

observation space, and $\pmb{R}$ is the observational error covariance matrix. In the case of a state vector of

emission fluxes, the observation operator $H(\cdot)$ is a CTM mapping emissions to concentrations. Solving

for the minimum of the cost function ( $\nabla J(\pmb{x}) = \pmb{0}$) defines the optimized posterior (also called analysis)

estimate $\pmb{x}^{\pmb{a}}$ for the state vector.

In the case where $H(\cdot)$ is linear (i.e., representable by a matrix), an analytic solution is available

with closed-form characterization of the posterior error covariance matrix [*Rodgers*, 2000]. In nonlinear



or high-dimensional linear cases, a variational approach can be used instead to iteratively minimize the

cost function by numerical methods. The three-dimensional variational approach can be used to calculate the gradient of the cost function in cases where observations $y$ fall close enough in time to the prior model state $x^b$ that time-evolution of the physical system can be neglected [*Asch et al.*, 2016]. Four-dimensional variational assimilation (4D-Var) accounts for nonlinear evolution of the system over the course of an assimilation time window through use of the adjoint of the physical model, which requires construction

of the tangent linear model (TLM) for the CTM; the TLM aligns the model state with observations in time while preserving the correct evolution of the physical system [*Courtier et al.*, 1994]. Like 4D-Var, the Kalman filter approach solves the cost function for the time evolution of the state vector by sequential assimilation of a time series of observations over successive assimilation time windows. The original Kalman filter requires a linear forward model, but it can be combined with the TLM of the physical model

to form the extended Kalman filter (EKF) which applies to nonlinear problems. The EKF has been used for atmospheric chemistry problems such as quantifying emissions of nitrogen oxides ($NO_x \equiv NO + NO_2$) from $NO_2$ satellite data [*Mijling and van der A*, 2012; *Ding et al.,* 2017]. All these methods require independent and often uncertain estimates of the prior and observational error covariance matrices $P^b$ and $R$.

Ensemble Kalman filters, including the localized ensemble transform Kalman filter (LETKF) used in this work [*Hunt et al.*, 2007], apply an ensemble of CTM simulations over successive assimilation time windows to approximate the prior error covariance matrix $P^b$ and its evolution over time. Like EKF and 4D-VAR, LETKF can be readily applied to nonlinear problems; however, it avoids the need for a TLM because it is powered by an ensemble of CTM simulations which capture the nonlinearity of the system.

Each ensemble member is initialized with random perturbations applied to emissions or concentrations of interest, and the ensemble is evolved for the assimilation time window using the CTM. At assimilation time, the ensemble spread is used to approximate the prior error covariance matrix $P^b$ and from there solve for the minimum of the cost function. The ensemble is then updated to reflect the optimized state, including emissions and concentrations, and the cycle repeats as in the case of the classic Kalman filter.

In practice, even though the state vector optimized is quite large, the ensemble can be of modest size



(typically 32 or 48 members) and the LETKF will converge on the correct solution as time progresses [*Hunt et al.*, 2007].

LETKF has been used extensively in chemical data assimilation, and has benefits compared with other algorithms, most notably the ease of implementation for a wide variety of simulations. LETKF and related ensemble Kalman filter methods have been used for $CO_2$ flux inversions [*Liu et al.*, 2016; *Kong et al.,* 2022], single-species studies of $NO_2$, $SO_2$, and $NH_3$ emissions [*Miyazaki et al.*, 2012a; *Dai et al.,* 2021; *van der Graaf et al.,* 2022], and analysis of methane emission trends [*Zhu et al.,* 2022]. Multi-species assimilation, 4D assimilation of temporally scattered observations, and flexibility in state vector definition are easy to implement under the LETKF framework; the algorithm also provides detailed error characterization including correlations as part of the solution. However, because ensemble methods rely on a relatively small number of simulations to simulate the problem space, the benefits of the LETKF come with issues of undersampling which will be discussed in section 2.2.

The ability of LETKF to simultaneously assimilate concentrations and emissions is of special importance to atmospheric chemists. In chemical data assimilation for operational forecasting, updates are often only applied to concentrations but this fails to address the root issue of incorrect emissions, an especially acute problem for species with short lifetimes such as $NO_x$ [*Inness et al.,* 2015]. On the other hand, inverse studies focused on optimizing emissions attribute all systematic discrepancies between the model and observations to emissions, even though CTM transport or observing errors may be responsible; indeed, $CO_2$ flux estimates calculated via inverse methods have been shown to be sensitive to transport errors [*Schuh et al.*, 2019; 2022]. Optimizing concentrations as well as emissions allows the data assimilation system to address both issues, assuming that prior error settings are posed appropriately. While this is possible to do with other algorithms, it is easy to do with LETKF due to the ability to add any additional parameter to the prior error covariance matrix and apply variable localization methods to optimize the application of observational constraints on different sets of concentrations and emissions.

Here we present the CHemistry and Emissions REanalysis Interface with Observations (CHEEREIO), a user-friendly tool that provides a platform for versatile LETKF chemical data assimilation powered by the widely-used GEOS-Chem CTM. Implemented as a lightweight wrapper for GEOS-Chem, CHEEREIO gives users the ability to design and run chemical data assimilation



applications without modifying model source code or learning a new codebase. CHEEREIO's flexibility
and simple design are enabled by the LETKF algorithm and the modular structure of GEOS-Chem, in
particular its HEMCO data input component [*Keller et al*., 2014; *Lin et al*, 2021]. CHEEREIO is designed
to be easily configurable for a range of applications including multi-species data assimilation, joint
optimization of emissions and concentrations, and near-real-time monitoring of emissions. Coded in
Python with an object-oriented framework, CHEEREIO readily accommodates new observation
operators such as for new satellite instruments. CHEEREIO and all of its components are open source,
ensuring scientific transparency. This paper provides a high-level overview and demonstration of
CHEEREIO; detailed documentation and user support are available online (cheereio.readthedocs.io).

## 2 CHEEREIO components

### 2.1 The physical model: GEOS-Chem

GEOS-Chem is a three-dimensional CTM driven by assimilated meteorological data from the Goddard
Earth Observation System (GEOS) of the NASA Global Modelling and Assimilation Office (GMAO),
either the GEOS Fast Processing (GEOS-FP) data at 0.25°x0.3125° native resolution or the GEOS
Modern-Era Retrospective Analysis for Research and Applications, version 2 (MERRA-2) data at
0.5°x0.625° native resolution, extending from the surface to the mesopause. It simulates atmospheric
species concentrations by solving the coupled 3-D Eulerian continuity equations on a global or user-
selected nested domain at the native grid resolution of the GEOS data or at degraded resolution for
computational economy. Input data files are regridded on the fly to user-specified resolution using the
Harmonized Emissions Component (HEMCO) [*Keller et al*., 2014; *Lin et al*., 2021]. CHEEREIO supports
all GEOS-Chem applications from version 13.0.0 and later including oxidant-aerosol chemistry, aerosol-
130 only, carbon gases, and mercury, implemented as either global or nested-grid simulations. GEOS-Chem
High Performance (GCHP) [*Eastham et al.,* 2018; *Martin et al.,* 2022], which uses distributed memory
rather than shared memory for parallelization, is not currently supported.

HEMCO is a critical GEOS-Chem module enabling the interface with CHEEREIO. It can apply
gridded scaling factors stored in netCDF files to any input field, such as emissions. This allows emissions



updates calculated by CHEEREIO to be seamlessly loaded into GEOS-Chem without modification of source code.

## 2.2 The data assimilation algorithm: Localized Ensemble Transform Kalman Filter (LETKF)

The LETKF algorithm optimizes a state vector of emissions and concentrations to minimize the cost function in Equation 1 [*Hunt et al.,* 2007]. We initialize $m$ ensemble members at time $t_o$ and run the
140 forward model (GEOS-Chem) in parallel for a user-specified time (termed the assimilation window) for each of these ensemble members. Ensemble members can be thought of as a Monte Carlo sample representing the spread of atmospheric conditions resulting from our uncertainty in prior emissions; each member represents the atmospheric conditions from a random emissions perturbation sampled from a user-specified PDF — before assimilation begins, the ensemble is run for a spinup period to ensure each
145 ensemble member has distinct atmospheric conditions. If emissions are known, the problem can also be set up by perturbing concentrations to represent prior uncertainty in the atmospheric state. Ensemble size for atmospheric chemistry is typically between 24 and 48, with the exact number determined by sensitivity testing, where the user identifies a size that balances error minimization with computational feasibility; in general, fewer ensemble members are required if there are fewer parameters to optimize [*Miyazaki et*
*al.,* 2012b; *Liu et al.,* 2019]. After the runs complete, we construct the state vectors $x_i^b$ representing the concentrations and/or emissions of interest for each ensemble member (indexed by $i$).

The LETKF algorithm, which we describe in the remainder of this section, is typically applied to very large state vectors, for which global optimization would be computationally prohibitive. The solution of *Hunt et al.* [2007] is to localize the calculation within a certain radius of the grid cell being optimized,
considering only observations within that radius. Localized state vectors are formed by concatenating emissions at a given grid cell with concentrations within a given radius, producing rolling variation between neighboring localized state vectors. Beyond reducing state vector size, this approach creates an embarrassingly parallel problem, where the cost function can be minimized independently for every localized state vector. The *Hunt et al.* [2007] localization approach also minimizes spurious correlations,
which emerge in ensemble approaches due to a limited sample size; because the Monte Carlo sample is far smaller than the dimensionality of the state vector, random points will be spuriously correlated in the



prior covariance matrix $\boldsymbol{P^b}$ encoded by the ensemble, leading to an incorrect assimilation increment. The spurious correlation problem is especially pronounced between distant grid cells where we would expect correlations to be near-zero, a problem eliminated by appropriate localization. The precise radius used for localization should be determined by the user via sensitivity tests, considering that longer-lived species require larger localization radii; indeed, within a single inversion, multiple localization radii can be used for different components of the state vector [*Miyazaki et al.*, 2012b]. For the remainder of the equations in this section, all vectors and matrices are localized and computations are performed in parallel.

To optimize the emissions and concentrations of a given grid cell, we construct the ensemble state vectors $\boldsymbol{x_i^b}$ using model data. From these prior state vectors the prior perturbation matrix $\boldsymbol{X^b}$ is formed from the *m* vector columns $\boldsymbol{X_i^b}$:

$$X_i^b = x_i^b - \overline{x^b}; \; \overline{x^b} = \frac{1}{m}\sum_{i=1}^{m} x_i^b \tag{2}$$

Here $\boldsymbol{X_i^b}$ represents the *i*th column of the *n* x *m* matrix $\boldsymbol{X^b}$ where *n* is the length of the state vector; each column of $\boldsymbol{X^b}$ consists of the state vector from an ensemble member minus the mean state vector. The prior covariance matrix $\boldsymbol{P^b}$ can be constructed by multiplying $\boldsymbol{X^b}$ with its transpose (specifically, $\boldsymbol{P^b} = (m-1)^{-1}\boldsymbol{X^b}(\boldsymbol{X^b})^T$) but this is not used directly in LETKF calculations.

The model predictions made during the assimilation window must be compared to observations. Hence we construct prior vectors of simulated observations $\boldsymbol{y_i^b}$ and a corresponding simulated observation perturbation matrix $\boldsymbol{Y^b}$ formed from the *m* vector columns $\boldsymbol{Y_i^b}$:

$$Y_i^b = y_i^b - \overline{y^b}; \; \; y_i^b = H(x_i^b); \; \; \overline{y^b} = \frac{1}{m}\sum_{i=1}^{m} y_i^b \tag{4}$$

4D-LETKF, the method used in CHEEREIO, constructs $\boldsymbol{Y_i^b}$ such that all simulated observations are timed to line up as close as possible with actual observations [*Hunt et al.*, 2007]. 3D assimilation, by contrast, only aligns observations in space but uses a single model state (in particular, the state at assimilation time) to construct $\boldsymbol{Y_i^b}$, leading to significant representation error. For 4D-LETKF, we load in model history files closest in time to the observation of interest and accept a modest representation error. Hence in practice we apply the operator $H(\cdot)$ to the forward model history, not to the state vector which represents the





model state at a specific point in time. Methods which make use of the TLM, like 4D-Var and EKF, avoid

temporal representation error due to the continuous ingestion of observations on the internal time step of

the TLM, but they require major time investment in TLM development and maintenance.

Computation of the cost function in **equation 1** involves inversion of the prior error covariance

matrix $P^b$ but this is not possible in the state space (of dimension $n$) because by construction $P^b$ is of rank

$m - 1$ (the columns of the $n \times m$ matrix $X^b$ sum to the 0 vector). Hence an posterior error covariance

matrix $P^a$ must be estimated in the $m - 1$ dimensional subspace $S$ spanned by the ensemble

perturbations, where the inverse is well-defined. The mathematics simplify by treating $X^b$ as a linear

transformation from some m-dimensional space $\tilde{S}$ to $S$, allowing us to redefine the cost function

optimization in $\tilde{S}$ where the relevant quantities are well-behaved [*Hunt et al.*, 2007]. The posterior error

covariance in $\tilde{S}$ noted with a tilde $\widetilde{P^a}$ is an $m \times m$ matrix computed as follows:

$$\widetilde{P^a} = \left( \frac{(m-1) \cdot I}{1 + \Delta} + \gamma (Y^b)^T R^{-1} Y^b \right)^{-1} \tag{5}$$

The full derivation of $\widetilde{P^a}$ is given in Hunt et al. [2007]. Here $I$ is the $m \times m$ identity matrix, $R$ is the

observational error covariance matrix, and $\gamma$ is a regularization constant set by the user. $\widetilde{P^a}$ plays the same

role in LETKF assimilation as the posterior covariance matrix $P^a$ plays in the classical Kalman filter,

connecting the state vector entries so that the calculated update is consistent with the internal correlations

of the system. The regularization constant effectively scales observational errors and is designed to

balance the weight given to observations within the present assimilation window in a data assimilation

problem. $\Delta$ is an inflation factor specified by the user, usually between 0 and 0.1, which accounts for

overconfidence in the assimilated ensemble; $\Delta$ does not affect the ensemble mean but it does increase the

ensemble spread, with larger values pushing ensemble members away from the ensemble mean. In

practice, ensemble spread can decrease with each assimilation cycle to values so small that the system is

no longer able to update (near infinite confidence is given to the prior term, so subsequent observations

carry no weight). Indeed, if $\gamma$ balances the weight given to observations within the present assimilation

window, $\Delta$ can be thought of as a term that balances the weight given to observations from all previous

assimilation windows.

The mean posterior state vector in the original space is then given by



$$\overline{x^a} = \overline{x^b} + \gamma X^b \widehat{P^a} (Y^b)^T R^{-1} (y - \overline{y^b}) \tag{6}$$

where $y$ is the vector of observations. The posterior perturbation matrix is then given by

$$X^a = X^b \left( (m-1) \widehat{P^a} \right)^{\frac{1}{2}} \tag{7}$$

From here, the new ensemble state vectors can be constructed by adding $\overline{x^a}$ back to each column of $X^a$. The LETKF gives error characterization from the assimilation; to obtain this, we need to transform $\widehat{P^a}$ back from the space $\tilde{S}$ to the original state space; since we defined $\tilde{S}$ with the linear transformation $X^b$, the posterior error covariance matrix is given by

$$P^a = X^b \widehat{P^a} (X^b)^T \tag{8}$$

With the ensemble updated and errors characterized, the ensemble can be evolved using GEOS-Chem for the next assimilation window. Importantly, the ensemble is not reinitialized for these new runs; the assimilated state of the previous assimilation window becomes the initial prior state of the next assimilation window. When the runs in the new assimilation window complete, the whole LETKF cycle begins again.

Many variations of the ensemble Kalman filter algorithms have been developed in the chemical data assimilation literature, each designed to better handle the behaviors of certain atmospheric constituents. For example, the Carbon Tracker $CH_4$ system handles the assimilation of long-lived $CH_4$ via a sliding-window approach, where emissions or fluxes from a given time period are estimated several times using a varying set of observations that evolve in time [*Peters et al.*, 2005; *Bruhwiler et al.*, 2014]. Similarly, the run-in-place method changes the behavior of the assimilation window to better handle gases like $CH_4$ [*Liu et al.*, 2019]. With run-in-place activated, the LETKF assimilation update is calculated using a long period of observations (e.g. 1 week) but the assimilation window is advanced forward for a smaller amount of time (e.g. 1 day). Run-in-place simulations thus maintain linear growth in posterior perturbations and allow the period where the assimilation update is calculated to experience the emissions adjustment, giving the system more time to correct assimilation errors. CHEEREIO supports many of these variations on the LETKF, as discussed in Section 3.





## 3 Description of the CHEEREIO platform

In this section, we describe the implementation of the LETKF in the CHEEREIO platform. We designed
this tool to ensure maximum scientific flexibility for a diverse user base, while maintaining an abstracted
interface to make the tool easy to use.

### 3.1 General workflow

**Figure 1** shows a schematic of the CHEEREIO workflow including initialization, spinup, sequential
physical model runs and LETKF assimilation, and postprocessing. Here we give a high-level overview
of how CHEEREIO can be customized and deployed for any chemical data assimilation applications with
GEOS-Chem. In subsequent sections, we will offer more detailed descriptions of the software design and
structure, omitting technical details provided in the web documentation (https://cheereio.readthedocs.io).



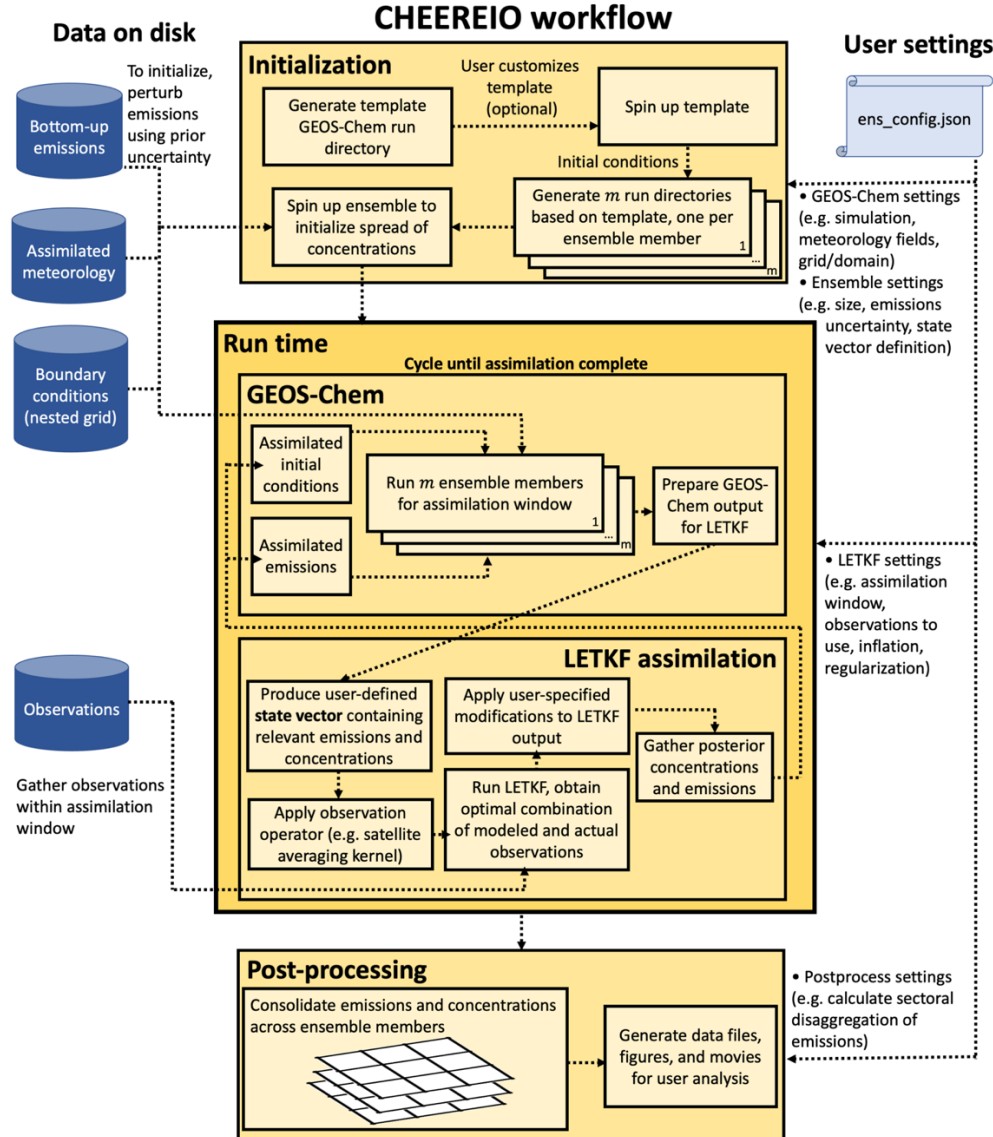

**Figure 1:** Schematic of the CHEEREIO workflow, divided into three steps: initialization, run time, and postprocessing. All simulations are initialized with a template GEOS-Chem run directory generated by CHEEREIO according to user-specified settings, which is then copied into an ensemble of *m* run directories, one per ensemble member, each with a unique set of emissions perturbed according to user settings. At run time, GEOS-Chem simulates atmospheric concentrations reflecting these perturbed emissions for each ensemble member, which are then compared to observations to generate an assimilated suite of concentrations and emissions via the LETKF procedure. After the specified period is assimilated, CHEEREIO postprocessing scripts consolidate the ensemble into a set of data files, figures, animations, and statistics for user analysis. Input data files are shown by dark blue cylinders at left, while user settings are shown at right.



The CHEEREIO software package includes a suite of shell and Python scripts for assimilation,
run management, observation operations, and postprocessing, which can be separated into three main
sequential periods in the CHEEREIO workflow, as shown in **Figure 1**: initialization time, run time, and
postprocessing time. Before initialization begins, users specify the simulation they would like to run via
an ensemble configuration file (ens_config.json). CHEEREIO generates a template GEOS-Chem run
directory based on user settings, which is copied into an ensemble of $m$ run directories, one per ensemble
member, each with a unique set of emissions perturbed according to user settings (**section 3.2).** After the
ensemble is initialized, the user submits a batch script which launches an ensemble of jobs, each running
an instance of GEOS-Chem. Once the model simulation for the assimilation window completes,
CHEEREIO gathers model output files and observation data and performs the LETKF assimilation. The
cycle of GEOS-Chem runs and LETKF assimilation repeats until the assimilation is complete for the
entire user-specified period. Run management and LETKF implementation are discussed in **section 3.3**.
Upon completion, the user can execute a postprocess workflow job to make a default set of figures,
movies, and consolidated data files; they can also deploy pre-written functions to produce custom output
and statistics (**section 3.4**). In the coming sections, we expand on each of these components of the
CHEEREIO workflow.

**3.2 Ensemble initialization**

The CHEEREIO ensemble initialization workflow is divided into four phases, as shown in **Figure 1**: (1)
template run directory creation; (2) spin up of template run directory; (3) ensemble initialization and prior
emissions sampling; and (4) spin up of the ensemble spread.

Initialization begins when the user specifies the simulation they would like to run by modifying a
configuration file (ens_config.json) which includes all model and assimilation settings. **Table 1** lists
important parameters that can be tuned in this configuration file; detailed instructions are in the online
documentation (cheereio.readthedocs.io). CHEEREIO then creates a template GEOS-Chem run directory
reflecting user settings, which will eventually be copied into an ensemble of $m$ run directories, one per
ensemble member; if users modify the template before the ensemble is created, such as to customize
emissions inventories, their adjustments will be reflected in each ensemble member. Hence the template



run directory allows users to customize their simulations beyond the parameters available in the ens_config.json configuration file. Like any atmospheric model, CHEEREIO must be spun up before any run begins so that it reflects realistic atmospheric conditions; spinning up the template run directory allows the user to run one universal spin up simulation for all *m* ensemble members.

**Table 1.** Selected parameters set by the CHEEREIO configuration file

| Parameter | Description |
| --- | --- |
| General parameters | |
| sim_name | type of GEOS-Chem simulation (oxidant-aerosol chemistry, aerosol-only, carbon species, mercury) |
| res | Horizontal resolution |
| region | For nested simulations, specifications for the nested domain |
| start_date, end_date | Start and end dates for the assimilation |
| burn_in_end | Discard results prior to this date |
| assim_time | Length of the assimilation window |
| State vector settings | |
| state_vector_conc | Species concentrations in state vector |
| control_vector_emis | 2-D emissions in state vector |
| state_vector_conc_representation | Representation of concentrations in the state vector (all 3D values, column sums, surface values, etc) |
| Output configuration | |
| HistorySpeciesConcToSave | Species concentrations to save to history files |
| HemcoDiagsToProcess | Which HEMCO diagnostics to include in postprocessing, such as total anthropogenic emissions of a given species. |
| Observation configuration | |
| observed_species | Species observed |
| OBSERVER_dirs | Directory storing observation files |
| LETKF settings | |
| regularizing_factor_gamma | Parameter $\gamma$, adjusts weight assigned to observations |



| inflation_factor | Ensemble inflation factor $\Delta$ |
| --- | --- |

With the template initialized, compiled, and spun up, CHEEREIO copies the template into an ensemble of $m$ run directories. Each ensemble member is differentiated by a unique initial perturbation to user-specified emissions, reflecting prior uncertainty in emissions. For example, users interested in assimilating $NO_2$ observations might specify that they have some prior uncertainty in $NO_x$ emissions;
CHEEREIO will then initialize unique grids of $NO_x$ emissions in each ensemble member run directory by drawing samples from a user-specified PDF that perturb existing emissions inventories. These prior errors can be sampled from a normal distribution and can include spatial correlations specified by a correlation distance.

Users can choose to use emissions sampled from either a normal or lognormal spread. If users opt
for lognormal emissions, then CHEEREIO samples multiplicative perturbations from a normal distribution centered on zero and then exponentiates to obtain a lognormally distributed sample with a mode of one (i.e. the prior). To meet the LETKF algorithm assumptions, in the lognormal case emissions are transformed back into a normal distribution during the LETKF calculation, before being exponentiated back to a lognormal for use in GEOS-Chem. Benefits of using a lognormal spread include a natural
protection against negative scaling factors (the lognormal distribution is positive) and a more realistic representation of uncertainties on emissions inventories.

CHEEREIO grants wide flexibility to users in how emissions perturbations are defined across the ensemble. Users can group emissions of multiple species together into one consolidated entry in the state vector (e.g. $NO_x$), updated at once at assimilation time. Users can also differentiate emissions by source
by separately perturbing subsets of emissions, such as methane from oil and gas and methane from agriculture. The resulting assimilation will provide the user separate emissions updates for each source, allowing users to easily run source attribution studies. Sectoral separation of emissions is implemented naturally in the LETKF formulation by defining the state vector so that separate source sectors have separate 2D representations; if sources overlap in space, the assimilation update will increment both
according to the correlation strength in the prior error covariance matrix, which the user must keep in mind while interpreting source separation results.



Before the assimilation cycle begins, users must run a CHEEREIO-specific spin up process to create a spread in simulated atmospheric conditions across ensemble members, reflecting the initial perturbations in emissions. Because the LETKF algorithm uses spreads in simulated concentrations across the ensemble to approximate the prior error covariance matrix $\boldsymbol{P^b}$, the model must be run for some period before assimilation begins in order to ensure that variations in concentrations across ensemble members reflects variations in emissions. If this ensemble-wide spin up is neglected or run for too short a period, $\boldsymbol{P^b}$ will be too small and observations will be neglected (because they will be weighted negligibly in the cost function).

**3.3 Runtime**

**Figure 2** shows a schematic of the CHEEREIO runtime processes. From a computational cluster perspective, CHEEREIO is an array of $m$ jobs, where $m$ is the number of ensemble members specified by the user; each job is allocated $p$ cores as specified by the user. Each job alternates between running GEOS-Chem for an ensemble member and running assimilation scripts for a subset of grid cells — each parallelized separately across the $p$ cores allocated to each job. In this section, we discuss the implementation and control of this complex of processes.





**Figure 2**: Schematic of CHEEREIO runtime routines and job control procedures. CHEEREIO is run as an array of *m* separate jobs on a computational cluster, one for each ensemble member. These *m* jobs, operating in parallel, alternate between running GEOS-Chem and running the LETKF algorithm for a subset of grid cells, as shown by the light yellow boxes; the *m* jobs are coordinated by a single job controller shared by the entire ensemble (shown in light red), ensuring that the ensemble remains synchronized. Boxes in blue show data input into CHEEREIO processes.

### 3.3.1 Job control

As shown in **Figure 2**, CHEEREIO begins when the user submits a job array initializing *m* jobs, one for each ensemble member, each consisting of *p* cores within a single node. Within each of the *m* jobs, the CHEEREIO runtime process is implemented as a shell loop that repeats until the user-specified period of interest is processed, switching smoothly between running GEOS-Chem and running LETKF assimilation



calculations. Because the LETKF algorithm is an embarrassingly parallel algorithm, there is no need for

complex cross-node parallelization schemes powered by the Message Passing Interface (MPI). Instead, each of the *m* jobs (parallelized independently across *p* cores) is coordinated by a job controller, which executes the processes shown in light red on **Figure 2**. The job controller synchronizes GEOS-Chem runs and LETKF assimilation routines across the ensemble, ensuring that all jobs remain connected to one another.

At the start of a given assimilation window, each of the *m* jobs calls GEOS-Chem for the current assimilation window. GEOS-Chem is parallelized within each job across *p* cores with OpenMP. After completing GEOS-Chem for the assimilation window, each individual job hangs until the job controller indicates that assimilation can begin. Once all GEOS-Chem runs are complete, the job controller initializes the LETKF routine. Each computational core within each job (a total of *mp* cores) is pre-

assigned a set of grid cells to assimilate, as the LETKF algorithm is embarrassingly parallel by grid cell. As a result, the LETKF can make use of multi-node parallelization without MPI; assimilated grid cells are written to a temporary directory, which will be used to update the entire ensemble once all *mp* cores finish the LETKF calculation. Internal parallelization of *p* cores within each of the *m* jobs is handled by GNU Parallel [*Tange,* 2018]. Once all expected grid cells are present, the job controller gathers

assimilated grid cell files, which represent assimilated concentrations and emissions, and overwrites GEOS-Chem restart files (representing initial concentrations) and emissions for each ensemble member. The job controller then cleans up temporary files, advances the time period of interest to the next assimilation window, and signals the job array to begin another GEOS-Chem run. If the entire period of interest is complete, then the job controller ends the job array. Different LETKF options, activated from

the configuration file, change the behavior of the job control scripts; for example, with run-in-place activated (**Section 2.2**), CHEEREIO computes the LETKF assimilation update using a long period of observations (e.g., 1 week) but advances the assimilation forward for a smaller amount of time (e.g., 1 day).

       CHEEREIO can easily handle emissions updates without GEOS-Chem source code modification

because of the HEMCO input module [*Keller et al.*, 2014; *Lin et al.*, 2021]. Emission updates are represented by a gridded set of scaling factors, initially randomized for each ensemble member in the





initialization process, which are present in each ensemble member run directory in gridded COARDS-compliant netCDF format. After each assimilation calculation, the file is updated by CHEEREIO to include the latest scaling factors and corresponding timestamp. HEMCO can read and regrid these latest
scaling factors on the fly, apply them to the emissions fields, and feed the scaled emissions directly into GEOS-Chem, enabling seamless interoperability across CHEEREIO runtime processes.

### 3.3.2 Assimilation computation

LETKF assimilation is implemented in CHEEREIO using a structure of nested Python objects, designed primarily to ensure that new observation operators can immediately plug into CHEEREIO and work
automatically, without requiring users to have deep knowledge of the CHEEREIO code structure. We use Python because of its familiarity to a broad user base, because of its ease of use, and because the object-oriented structure of the language makes it well suited to the modular design of CHEEREIO.

CHEEREIO works by creating a suite of objects called translators, which load data from gridded netCDF files used by GEOS-Chem runs, form one-dimensional ensemble state vectors $x_i^b$ and prior
vectors of simulated observations $y_i^b$ that are acceptable to the CHEEREIO LETKF routine, and convert assimilated state vectors back into a format acceptable to HEMCO for input into GEOS-Chem. Translator objects are assembled in a nested structure, with low-level translators performing IO operations and basic calculations to form vectors like $x_i^b$ and $y_i^b$, which are then passed to objects that operate at a higher level of abstraction. Abstract objects do the actual LETKF calculations without any knowledge of the GEOS-
Chem simulation or even the user-defined rules on how to construct the state vector, enabled by the fully general nature of the LETKF. Because all the details of a specific simulation are handled by low-level translators, which are designed to easily expand to include new capabilities added by the community, users are able to modify only one small part of CHEEREIO without compromising the overall workflow.

For example, CHEEREIO handles observations by using objects inheriting from the
390 Observation_Translator class, a low-level translator which loads observations from file and compares them to GEOS-Chem output. In object-oriented programming, inheritance can be thought of as a sophisticated form of templating. Indeed, the Observation_Translator class itself is mostly empty, and contains instructions to the user on how to write two standardized methods to (1) read observations from



file and process them into a Python dictionary formatted for CHEEREIO, and (2) generate simulated observations $y_i^b$ from GEOS-Chem output. Users can easily write their own class inheriting from Observation_Translator for a specific use case (like a particular surface or satellite instrument) by implementing these two methods, optionally employing a provided observation toolkit. Any class written with this strict template will then plug in automatically to the rest of the CHEEREIO workflow and can be activated from the main configuration file. CHEEREIO also comes with some pre-written observation operators (such as for the TROPOMI and OMI satellite instruments). Many different observation operators can be used simultaneously, making it straightforward to perform multispecies data assimilation or assimilation using both surface and satellite data within the CHEEREIO framework. Again, because Observation_Translators handle the details of interpreting a specific observation type, the rest of CHEEREIO can remain ignorant of specifics and operate in a fully abstract environment that can be reused for all simulations.

The Observation_Translator template includes tools that support aggregating observations into "super-observations" [*Eskes et al.*, 2003; *Miyazaki et al.*, 2012a]. If super-observations are enabled, CHEEREIO will average observations onto the GEOS-Chem spatiotemporal grid. Users can opt to supply a relative or absolute error for observations and opt to either (1) apply these values consistently regardless of whether observations are aggregated, or (2) reduce errors as observations are aggregated following a square root law or another functional form supplied by the user (such as an empirical curve) to account for correlations and model transport error. Users can also use error statistics supplied with the observations (such as retrieval errors), with the super-observation error standard deviation calculated according to a function they specify. The default super-observation error standard deviation $\sigma_{\text{super}}$ is calculated as follows:

$$\sigma_{\text{super}} = \sqrt{\left[\left(\frac{1}{n}\sum_{i=1}^{n}\sigma_i\right)\cdot\left(\frac{1-c}{n}+c\right)\right]^2 + \sigma_{\text{transport}}^2} \tag{9}$$

Here $\sigma_i$ is an individual observation error standard deviation (in the same units as the observation), $n$ is the number of observations aggregated into a super-observation, $c$ is the error correlation between the individual observations averaged into the super-observation, and $\sigma_{\text{transport}}$ represents model transport





errors that can be supplied by the user. Model transport error is included as a separate term because
transport errors are perfectly correlated for a given model grid cell and therefore irreducible by averaging.

## 3.4 Postprocessing

When the CHEEREIO runtime process completes, users can execute the postprocess batch script to automatically consolidate GEOS-Chem diagnostic and emissions output into netCDF files, along with a file pairing actual observations with simulated observations from the ensemble. Users can also run a
control ("prior") simulation with no assimilation within the CHEEREIO environment; output from this run is automatically handled by the postprocessing utility to produce plots and data that compare assimilated output to control output. CHEEREIO also produces a suite of graphs and animations depicting a variety of output including scaling factors, concentrations, emissions, and observation information. All plots of results in **Section 4** of this paper are generated by the CHEEREIO postprocessing
utility with no additional code. To facilitate additional analysis, a postprocessing toolkit is provided for user processing of both consolidated output files and raw ensemble output.

## 4 Example application: global optimization of methane emissions

Here we demonstrate an end-to-end example application of CHEEREIO to the problem of optimizing global emissions of methane with high temporal (weekly) resolution by assimilation of TROPOMI
satellite observations for the full year of 2019. The application uses the standard CHEEREIO configuration files, and all figures and statistics are automatically produced by CHEEREIO with no additional programming. There are weaknesses in the inversion parameters that we identify but do not try to resolve as the application is for demonstration purposes only.

## 4.1 Demonstration simulation setup

For our demonstration, we use the methane simulation from GEOS-Chem version 14.0.2 (doi: 10.5281/zenodo.7383492) at 2.0°x2.5° spatial resolution. For meteorology, prior emissions, and loss terms we use the default GEOS-Chem inventories and follow the setup described by *Qu et al.* [2021]. Methane emissions come from anthropogenic sources including livestock, oil and gas, coal mining,



landfills, wastewater, and rice cultivation, and from natural sources including wetlands and termites. Loss
is primarily due to oxidation by OH, with additional minor terms from oxidation by Cl atoms,
stratospheric oxidation, and soil uptake.

Methane observations used for data assimilation are from the TROPOspheric Monitoring
Instrument (TROPOMI) scientific product 2.2.0, shown in **Figure 3** [*Lorente et al*, 2021]. TROPOMI
retrieves daily global dry methane column mixing ratios ($XCH_4$) at 5.5x7 $km^2$ nadir pixel resolution at
~13:30 local solar time. In our demonstration, we filter TROPOMI observations to include only those
over land below 60 degrees latitude with a quality assurance value > 0.5; short wave infrared (SWIR)
albedo between 0.05 and 0.4 to avoid biases over dark scenes or highly reflective (often desert) scenes; a
low blended albedo (< 0.75) to avoid snow-covered scenes [*Wunch et al.*, 2011; *Lorente et al.*, 2021]; and
SWIR aerosol optical thickness less than 0.1. As shown in **Figure 3**, retrieval count after filtering varies
strongly by location; CHEEREIO regrids the native $XCH_4$ observations on-the-fly to the GEOS-Chem
grid resolution, as discussed later in this section.

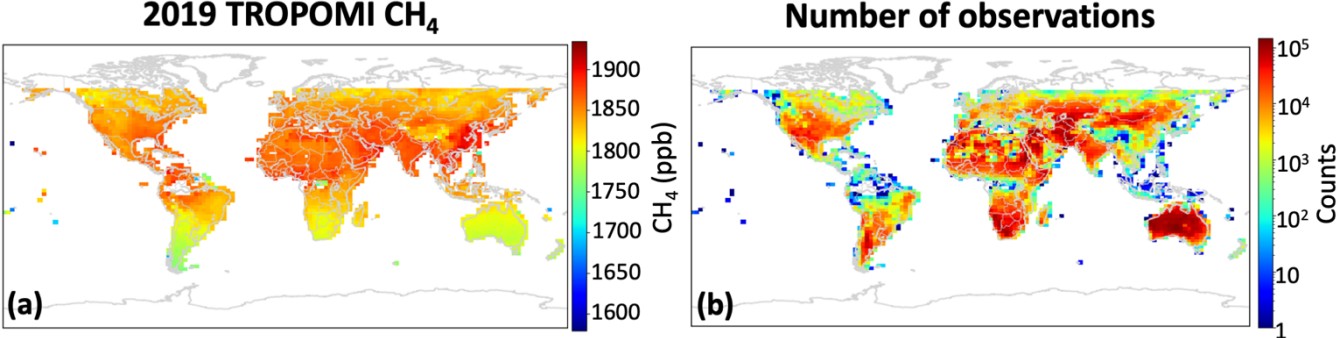

**Figure 3:** TROPOMI observations used in CHEEREIO demo for weekly inversion of methane emissions.
(a) Average TROPOMI $XCH_4$ for 2019, after filtering as described in the text. (b) Number of TROPOMI
observations used. Values are plotted on the GEOS-Chem 2°x2.5° grid.

A subset of the assimilation settings for this demonstration, passed to CHEEREIO through the
configuration file, are supplied in **Table 2**. The state vector includes weekly time-dependent global 3-D
concentrations as well as emissions on the 2°x2.5° grid over land excluding poleward of 60°. We use a
24-member ensemble, consistent with the LETKF ensemble size used for carbon fluxes in *Liu et al.*
[2019]. Each ensemble member is initialized with randomized methane emissions on the 2°x2.5° grid that
range from approximately 50% to 150% of prior values, based on a user-specified prior error parameter.



Initial emissions for individual members are sampled from a normal distribution with spatial correlation and are normalized so that the initial ensemble mean emissions equal the prior emissions on the 2.0×2.5º grid. We spin up the model for each ensemble member for four months with these initial emissions, and then further multiplicatively increase the ensemble standard deviation of methane concentrations by a factor of five; the goal of this scaling is to emulate a much longer spin up run of GEOS-Chem. We then adjust each ensemble member by the same global multiplicative factor so that the ensemble mean methane concentrations are equal to TROPOMI observations at the start of the assimilation period. Furthermore, we discard the first two months of assimilated output; we find that the LETKF system has a lag time between when assimilation begins (November 2018) and when emissions updates begin to stabilize, which we call the burn-in period. To reduce the time required for burn-in, for November and December 2018 we use a high regularization constant of $\gamma = 5$ to artificially increase the weight of observations during the burn-in period.

The user does not specify how assimilation increments are split between emissions and concentrations: the LETKF formalism simultaneously updates different aspects of the state vector, emissions and concentrations included, solely according to the correlations between state vector elements represented in prior error covariance matrix $\boldsymbol{P^b}$, which is determined by the spread of the CTM ensemble. The prior error in concentrations is determined by the spread in concentrations resulting from the perturbed emissions in each ensemble member. Nevertheless, performance can be enhanced by establishing different parameters for different components of the state vector, such as using different localization radii or inflation schemes [*Miyazaki et al.,* 2012b; *Bisht et al*., 2023].

**Table 2.** Selected parameters from CHEEREIO configuration file for methane demonstration

| Parameter[a] | Value |
|---|---|
| General parameters | |
| sim_name | CH4 |
| res | 2.0°x2.5° |
| start_date, end_date | 20181101, 20200101 |
| burn_in_end | 20181225 |




| | |
|---|---|
| assim_time | 168[b] |
| State vector settings | |
| state_vector_conc | CH4 |
| control_vector_emis | CH4 |
| state_vector_conc_representation | 3D |
| Output configuration | |
| HemcoDiagsToProcess | EmisCH4_Total |
| Observation configuration | |
| observed_species[c] | CH4_TROPOMI : CH4 |
| TROPOMI_dirs[c] | CH4 : /path/to/tropomi/netcdf/files |
| LETKF settings | |
| regularizing_factor_gamma | 1 |
| inflation_factor | 0.03 |

[a]Parameter descriptions are in **Table 1**
[b]Hours, equal to 1 week
[c]Many parameters are supplied in key:value form; details in the online documentation

Following ensemble spinup and the burn-in months, we run the model with assimilation for one year (2019). We simultaneously assimilate 3D concentrations of methane as well as emissions; the LETKF algorithm natively computes prior error variance from the ensemble spread (for example, accounting for strong error correlation in the vertical). We use an assimilation period of one week and

optimize grid cells following a horizontal localization radius of 500 km. We use an inflation factor $\Delta = 0.03$ and a regularization constant $\gamma = 1$. Moreover, we impose a zero floor on emissions; a lognormal emissions spread (**Section 3.2**) would be a better way to prevent negative emissions [*Maasakkers et al.*, 2019]. We aggregate TROPOMI methane observations into "super-observations" on the GEOS-Chem 2.0°x2.5° grid, reducing errors following equation 9, with individual observation error $\sigma_i = 17$ ppb,

transport error $\sigma_{\text{transport}} = 6.1$ ppb, and error correlation $c = 0.28$, values determined empirically for TROPOMI methane by *Chen et. al.* [2023].



## 4.2 Posterior solution and evaluation

**Figure 4** shows the adjustment of global methane concentrations (left) and emissions (right) from the assimilation calculation relative to the prior emission inventory used in the Control simulation. The

Control simulation produces methane concentrations slightly higher than observed by TROPOMI in January-June, leading to a downward correction of emissions. In July the situation reverses sharply as the Control simulation falls well below TROPOMI observations, likely because of seasonal underestimate of prior emissions from boreal wetlands and rice cultivation [*Maasakkers et al.,* 2019]. The assimilation responds with increased emissions but with a 1-month time lag reflecting the need to accumulate sufficient

observations to inform the state vector. CHEEREIO's run-in-place capability (**Section 3.3.1**) would allow the LETKF algorithm to mitigate this lag, as would a sliding-window approach such as that used by the Carbon Tracker $CH_4$ system [*Bruhwiler et al.*, 2014; *Liu et al.*, 2019]. Additional improvements could come from optimizing column $CH_4$ rather than 3-D concentrations, which would make the system more robust against non-emissions errors such as transport.

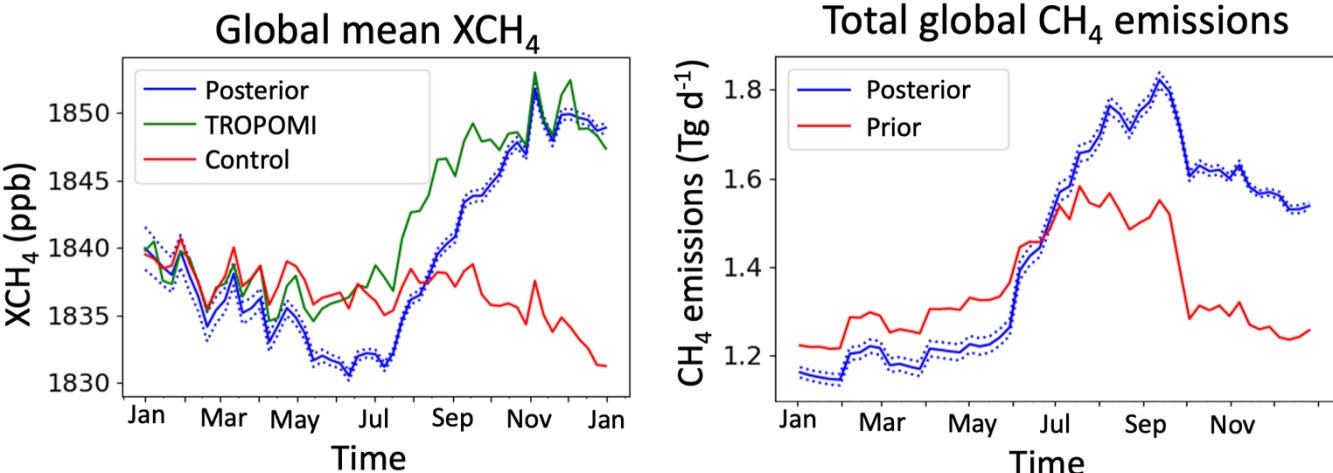

**Figure 4:** Time-dependent corrections to global methane concentrations and emissions from the weekly LETKF assimilation of TROPOMI observations as demonstrated by CHEEREIO. The left panel shows the global mean methane dry column mixing ratios (XCH4) in the TROPOMI observations, the Control simulation using prior emissions, and the simulation using posterior emissions. The right panel shows the

global prior and posterior emissions. Posterior values are ensemble means from the assimilation (standard deviations from ensemble members as dotted lines). The assimilation was conducted for one year from January through December 2019.



**Figure 5** shows the prior and posterior emissions for December 2019, along with the posterior error standard deviation. **Figure 6** evaluates the ability of the posterior simulation to better fit the
525 TROPOMI observations in that same month. Model bias is reduced by the LETKF assimilation procedure, with a mean bias of $1.2 \pm 10.6$ ppb in the ensemble (assimilated) mean as compared to $-16.1 \pm 12.3$ ppb in the prior (no assimilation) simulation.

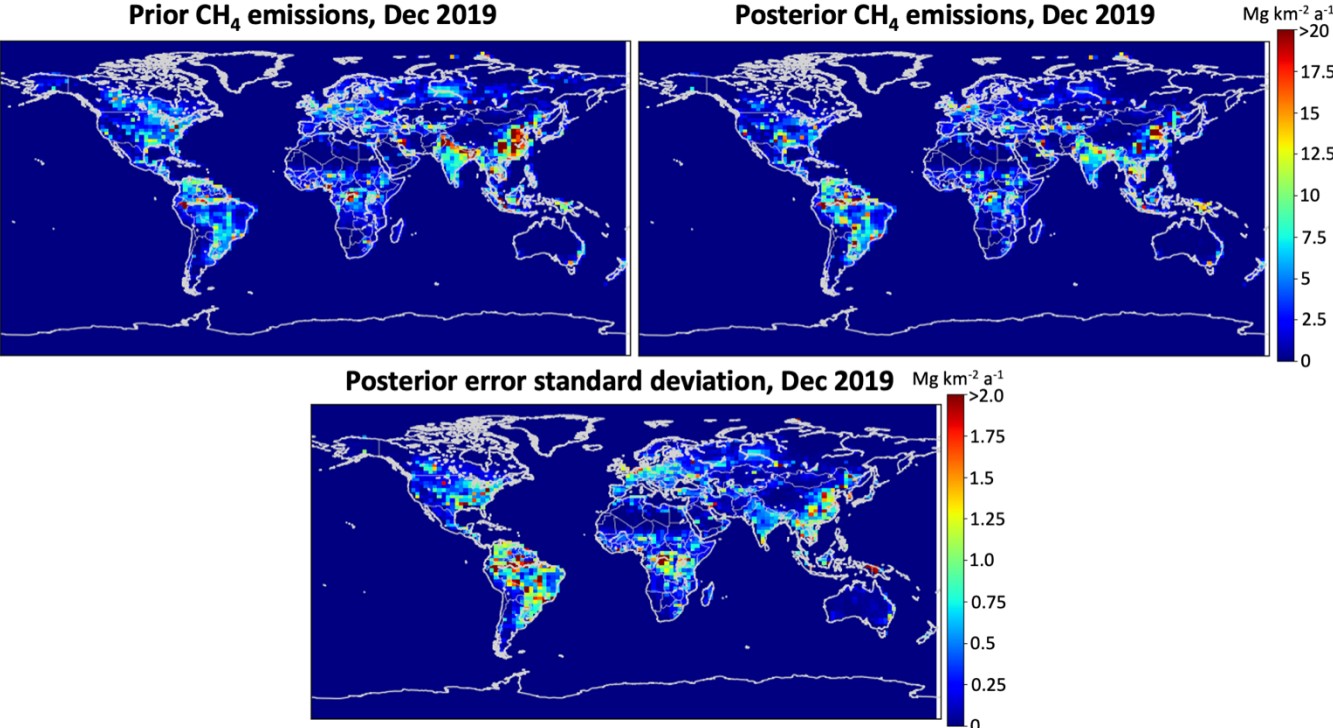

**Figure 5**: Prior and posterior estimates of methane emissions in December 2019, and error standard deviations on the posterior estimates. The posterior estimates are the means of the 24-member ensemble and the posterior error standard deviations are defined by the spread in the ensemble.





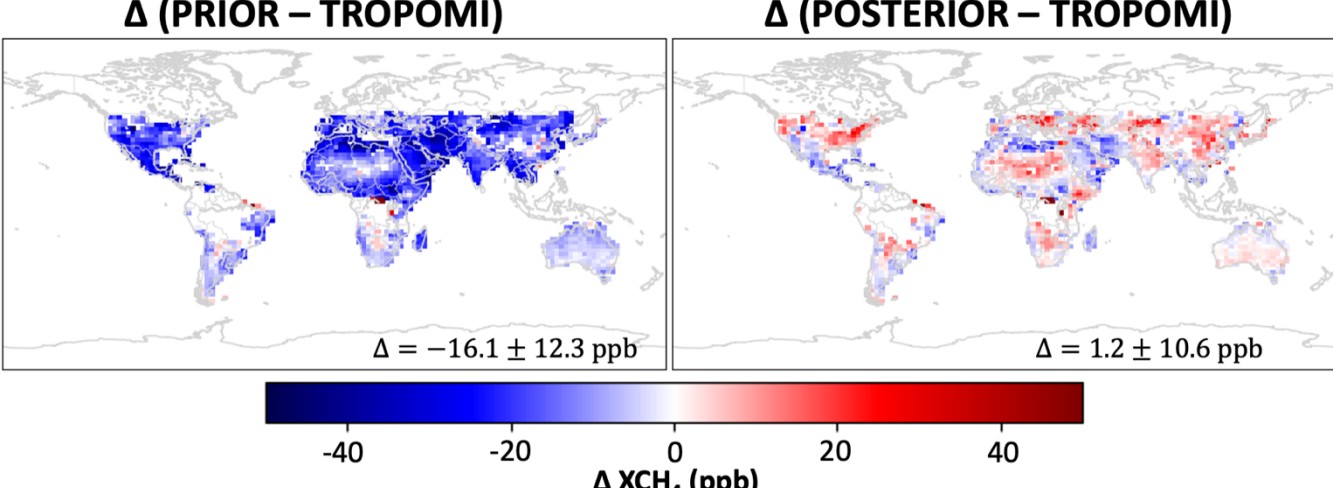

**Figure 6:** Comparison of simulated dry column mixing ratios (XCH4) with prior or posterior emissions to TROPOMI observations for December 2019. Values are monthly mean differences between the simulated XCH4 (with TROPOMI observation operator applied) and the TROPOMI observations. Mean bias and standard deviation are given inset.

The use of CHEEREIO to optimize methane emissions represents a substantial improvement in computational performance relative to an analytical inversion approach. *Qu et al.,* [2021] previously applied the analytical approach with GEOS-Chem to optimize global methane emissions at 2°x2.5° resolution for 2019. Their formulation had 4190 state vector elements, which required a total of 4190 perturbed GEOS-Chem simulations. By contrast, our approach only required 24 GEOS-Chem simulations to form the ensemble. Within CHEEREIO, the model spent an average of 29.8% of wall time running GEOS-Chem and 70.2% running LETKF routines. The relatively high overhead of LETKF routines appears in part because the methane GEOS-Chem simulations are relatively fast; full oxidant-aerosol chemistry simulations are considerably more expensive but will have a similar LETKF overhead cost. Accounting for the relatively high LETKF overhead of CHEEREIO, we achieve a factor of 52x reduction in computational costs relative to an equivalent analytical inversion.

## 5 Conclusions and future development

We presented the CHemistry and Emissions REanalysis Interface with Observations (CHEEREIO), a user-friendly Python-based tool that supports localized ensemble transform Kalman filter (LETKF)





chemical data assimilation (including emissions inversion) powered by the GEOS-Chem chemical transport model. CHEEREIO provides application-ready and versatile software for users to exploit observations of atmospheric composition from satellites and other platforms to infer emissions and optimize 3-D concentration fields, including error characterization. The CHEEREIO source code is available for download at https://github.com/drewpendergrass/CHEEREIO and is documented at https://cheereio.readthedocs.io.

We choose the LETKF algorithm because of its general applicability for linear and nonlinear problems, multiple observational data streams, flexible state vector definition, and error characterization of the solution. Its ensemble-based structure is well suited to developing a simple but powerful tool that requires neither the forward model adjoint nor modifications to model source code, and that can be run on supercomputing clusters as an embarrassingly parallel task. Use of GEOS-Chem as forward model allows a wide range of applications to tropospheric and stratospheric chemistry, as well as simpler linear problems (such as $CO_2$ or methane inversions), on regional scales with spatial resolution down to 25 km (native resolution of GEOS-Chem) as well as global scales. A critical component of GEOS-Chem is its data input module HEMCO, which allows emissions updates from the assimilation steps to pass seamlessly to GEOS-Chem without code modification.

We designed CHEEREIO so that users can specify their data assimilation problem through a basic configuration file expressing the state vector to be optimized, the prior information, the GEOS-Chem specifications (type of simulation, resolution, assimilation period), and the LETKF parameter information. LETKF implementation is handled under the hood by a suite of CHEEREIO scripts that do not require user familiarity. Users can readily add new observation operators as needed without modifying the rest of the CHEEREIO code base.

We demonstrated CHEEREIO's ability with an example application of assimilating concentrations and emissions of atmospheric methane for one full year using observations from the TROPOMI satellite instrument. The entire demonstration was run out-of-the-box, with no additional coding beyond the base CHEEREIO code. Output figures and statistics presented here were auto-generated by the CHEEREIO postprocessing utility. Accounting for the relatively high overhead of the LETKF computation in the methane case, our approach represents a factor of 52 reduction in



computational cost relative to an equivalent analytical inversion. Because of computational cost savings, we envision CHEEREIO's methane data assimilation can serve as a global complement to the regional nested-grid simulations offered by the Integrated Methane Inversion (IMI), a similar software platform designed for analytical methane inversions [*Varon et al.,* 2022]

585   More work can be done to improve CHEEREIO and expand its capability. Although CHEEREIO is designed as a lightweight software wrapper that is accessible to the GEOS-Chem community, future development will incorporate software components from the Joint Effort for Data Assimilation Integration (JEDI), a C++ and Fortran-based platform for model-generic data assimilation [Trémolet and Auligné, 2020]. In particular, we plan to support observation operators implemented as part of the JEDI

590 Unified Forward Operator (UFO) initiative, allowing users to leverage the wide library of instruments supported by JEDI without duplicating code themselves. The LETKF algorithm is agnostic to the forward model, making it practical in theory to use any chemical transport model as forward model for CHEEREIO. In practice, models that use HEMCO for emissions input would be easiest to support. The NASA GEOS and NCAR CESM Earth system models have adopted HEMCO [*Lin et al*., 2021], and the

595 LETKF approach implemented in CHEEREIO would allow optimization of emissions as part of chemical data assimilation in these models. Because CHEEREIO is designed to take advantage of the embarrassingly-parallel LETKF algorithm without using shared memory, it is reasonably straightforward to extend the system to models parallelized with MPI such as GCHP. Further improvements to the LETKF parallelization routine, in particular methods to share memory resources within Python, can also be

600 applied to reduce I/O overhead, reduce memory use, and improve assimilation wall time. CHEEREIO can be ported on the cloud, taking advantage of GEOS-Chem and satellite data already hosted there [*Zhuang et al*., 2019, 2020; *Varon et al*., 2022], thus bringing compute capacity to big data rather than requiring cumbersome data downloads. Cloud implementation would facilitate the development of near-real-time chemical data assimilation products for emissions monitoring and air quality forecasts.

605 **Code availability**

The CHEEREIO 1.0 source code is available at https://github.com/drewpendergrass/CHEEREIO and is documented at https://cheereio.readthedocs.io. The version of CHEEREIO used in this paper is archived

at doi.org/10.5281/zenodo.7781437. GEOS-Chem version 14.0.2 source code is archived at doi.org/10.5281/zenodo.7383492.

**Data availability**

The CHEEREIO model output from the demonstration section of the paper is available at https://doi.org/10.5281/zenodo.7806312, and contains all necessary data for reproducing figures 3-6 including prior methane emissions, posterior methane emissions, and TROPOMI XCH$_4$ paired with simulated prior and posterior GEOS-Chem XCH$_4$. The raw TROPOMI science data fed into CHEEREIO is available from SRON (https://ftp.sron.nl/open-access-data-2/TROPOMI/tropomi/ch4/, last accessed April 6, 2023) or on request.

**Author contributions**

DCP and DJJ contributed to the study conceptualization. DCP developed the model code, with contributions from DJV, HN, and MS. KM, KWB, DJJ, and DCP contributed to the methods' development. DCP performed the data analysis. DCP wrote the original draft, and all authors reviewed and edited the manuscript.

**Acknowledgements**

This research has been supported by the National Aeronautics and Space Administration (NASA; Carbon Monitoring System; grant no. 80NSSC21K1057). DCP was funded by an NSF Graduate Research Fellowship Program (GRFP) grant. A part of the research was conducted at the Jet Propulsion Laboratory, California Institute of Technology, under a contract with NASA.



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
