# Peer review of "CHEEREIO 1.0: a versatile and user-friendly ensemble-based chemical data assimilation and emissions inversion platform for the GEOS-Chem chemical transport model"

_EGUsphere, 2023_

## Author Response (AR1)

**Response to reviewer 1**.

In this paper, the authors introduce CHEEREIO 1.0, a novel chemical data assimilation toolkit. This toolkit is designed to optimize emissions and concentrations of chemical species based on atmospheric observations from satellites or suborbital platforms. The toolkit utilizes the GEOS-Chem chemical transport model and a localized ensemble transform Kalman filter algorithm to determine the Bayesian optimal emissions and/or concentrations of a set of species. The Python-based design of CHEEREIO is commendable for its user-friendliness, allowing users to easily add new observation operators such as for satellites.

The toolkit demonstrates impressive computational performance, achieving a 50-fold improvement compared to the equivalent analytical inversion of TROPOMI observations. This advancement holds significant potential to enhance the efficiency of atmospheric chemistry research.

The paper is well-structured, with the authors providing a clear explanation of the methodology used.

- Response: We thank the reviewer for their helpful feedback, which has clarified the methodology in the manuscript and offered potential directions for future research.

Minor Comments:

1. Around line 540, the authors compare the computational efficiency of the toolkit with the analytical inversion. While the superior computational efficiency of the toolkit is evident, the comparison seems incomplete without a discussion on inversion accuracy. I would recommend the authors to include a comparison of the inversion accuracy.

- Response: In lines 574-81, we add a quantitative comparison with the posterior emissions and OH adjustment of *Qu et al*., [2021], concluding that the simulations are globally consistent with some regional differences. We briefly discuss possible reasons for posterior differences and indicate that further comparison between these methodologies would be of interest in future work.

2. The reduction in computational cost is mainly achieved by using a smaller ensemble size, primarily through the application of localization. It would be interesting to know if using the full rank ensemble size (as in the analytic solution) provides a superior solution. Also, can the analytical inversion also apply localization?

- Response: We clarify in lines 160-3 that ensemble approaches exhibit "diminishing returns" as additional simulations are added, whereas the number of simulations required for the analytical approach is set by the size of the state vector. Additionally, we offer a short explanation for why analytical inversions do not need to apply localization in lines 179-80.

3. The authors briefly mention the limitations of CTM bias/error. However, the described CHEEREIO system does not seem to address errors in the model, such as perturbations in meteorological condition and physics. It would be beneficial if the authors could elaborate on how these potential sources of error are addressed in the CHEEREIO system.

- Response: We add an explanation in lines 444-8 that users can represent model transport errors by using the residual error method and equation 8 in the paper, although this is unable to correct systematic bias from meteorology or chemistry. We indicate that future expansion of CHEEREIO to coupled chemistry-weather models would fully address the CTM limitations the reviewer raises.

**Response to reviewer 2**.

The authors developed a user-friendly chemical data assimilation platform based on the GEOS-Chem model and the LETKF algorithm, which can optimize emissions and concentrations of different species constrained by satellite observations. The GEOS-Chem model is a powerful tool for atmospheric modeling widely used in the scientific community and the LETKF algorithm has been proven to work efficiently for both linear and nonlinear problems. I feel that the inversion toolkit developed in this paper is very important and useful, and I believe that this tool will promote the research on chemical data assimilation and emission inversions. This manuscript is well-structured and well-written. I suggest a minor revision after addressing my comments below.

- Response: We thank the reviewer for their feedback, which has better contextualized the results and offered a more intuitive view of the algorithm behavior to the reader.

1. There are already a few open-source inversion tools that have been developed. I suggest that the authors briefly compare CHEEREIO and other platforms and discuss the differences and advantages of this new platform.

- Response: We added a discussion of representative open-source inversion tools in lines 121-6 and contextualized CHEEREIO's contributions.

2. I understand that due to length limitations, the authors provide detailed documentation of CHEEREIO online (cheereio.readthedocs.io), instead of presenting them within the manuscript. If possible, I suggest the authors explain a little bit more of the key parameters and settings of CHEEREIO, which could largely influence the emission inversion results. This can help the readers quickly understand the most important parts of the LETKF algorithm, as well as the most sensitive parameters that should be concerned about.

- Response: We added the localization radius as a key parameter to Table 1, and offered a brief overview of the most critical parameters in lines 294-9.

3. In Sect. 4, the authors mentioned a comparison between the example application of $CH_4$ emissions inversion with Qu et al. (2021), which optimized global $CH_4$ emissions at the same horizontal resolution using the analytical approach. Is it possible to directly compare the

posterior CH$_4$ emissions with Qu et al. (2021) in the main text? This would be helpful to understand the differences between the LETKF algorithm and the analytical approach.

- Response: As mentioned in our response to Reviewer 1, in lines 574-81, we add a quantitative comparison with the posterior emissions and OH adjustment of *Qu et al.*, [2021], concluding that the simulations are globally consistent with some regional differences. We briefly discuss possible reasons for posterior differences and indicate that further comparison between these methodologies would be of interest in future work.